# Effect of Filtration Process on Oxidative Stability and Minor Compounds of the Cold-Pressed Hempseed Oil during Storage

**DOI:** 10.3390/antiox12061231

**Published:** 2023-06-07

**Authors:** Vincenzo Lo Turco, Federica Litrenta, Vincenzo Nava, Ambrogina Albergamo, Rossana Rando, Giovanni Bartolomeo, Angela Giorgia Potortì, Giuseppa Di Bella

**Affiliations:** 1Department of Biomedical, Dental, Morphological and Functional Images Sciences (BIOMORF), University of Messina, Viale Annunziata, 98100 Messina, Italy; vloturco@unime.it (V.L.T.); felitrenta@unime.it (F.L.); rrando@unime.it (R.R.); agpotorti@unime.it (A.G.P.); gdibella@unime.it (G.D.B.); 2Department of Veterinary Sciences (SCIVET), University of Messina, Viale Annunziata, 98100 Messina, Italy; 3Science4Life Srl, Viale Annunziata, 98100 Messina, Italy; gbartolomeo@unime.it

**Keywords:** hempseed oil, cold-pressed oils, oxidative stability, minor compounds, filtration process, storage

## Abstract

Cold-pressed hempseed oil (HO) has been increasingly exploited in the human diet for its excellent nutritional and healthy properties. However, it has a high content of polyunsaturated fatty acids (PUFAs) and chlorophylls, which inevitably accelerate its oxidative deterioration, especially in the presence of light. In this scenario, the filtration technology may ameliorate the oxidative stability of the oil, with positive effects on its nutritional quality and shelf life. Therefore, in this study, the oxidative stability and minor compounds of non-filtered and filtered HO (NF-HO and F-HO) were monitored over 12 weeks of storage in transparent glass bottles. F-HO showed a better hydrolytic and oxidative status than NF-HO during storage. As a result, F-HO also displayed better preservation of total MUFAs and PUFAs in the autoxidation process. Filtration consistently reduced chlorophylls, thus causing a variation in the natural color of HO. Accordingly, F-HO not only revealed an increased resistance to photooxidation but it was also suitable for storage in clear bottles within 12 weeks. F-HO predictably showed lower carotenoids, tocopherols, polyphenols, and squalene compared to NF-HO. However, filtration appeared to play a “protective role” toward these antioxidants, which had lower degradation rates in F-HO than NF-HO for 12 weeks. Interestingly, the element profile of HO was not affected by filtration and remained stable during the study period. Overall, this study may be of practical use to both producers and marketers of cold-pressed HO.

## 1. Introduction

Cannabis (Cannabaceae) is an annual flowering plant genus whose taxonomy has always been quite complex and troublesome. According to the most accepted interpretation proposed by Small and Cronquist [1], the genus is monospecific (*Cannabis sativa* L.) and includes two subspecies (*Cannabis sativa* L. subsp. sativa and *Cannabis sativa* L. subsp. indica (Lam.)), which are in turn characterized by several varieties, conferring relevant variability to the genus [2,3]. Such genetic diversity has played a key role in the long history of domestication of *C. sativa* L. and has contributed to providing the crop with special traits such as great versatility, environmental sustainability, prompt adaptation to different environmental conditions, and application in several fields [4]. In such a complex scenario, the main discrimination factor among the varieties is the content of the psychoactive molecule (-)-trans-Δ9-tetrahydrocannabinol (Δ9-THC, or simply THC), which allows one to discriminate the drug-type specimens (i.e., marijuana) from the fiber-type plants (i.e., industrial hemp) [5]. Whereas the market has notoriously focused on industrial hemp to produce fiber and hurds for various applications (e.g., textiles, paper, green building, bioplastics, bioenergy, etc.) [6], hempseeds—whose estimated production is of 6000 t per year in the EU [7]—have been undervalued and treated as a waste product [7,8]. However, hempseeds are a valuable source of essential fatty acids, minerals, dietary fiber, vitamins, phenolics, and, not least, essential aminoacids contained in the highly digestible proteins edestin and albumin [8,9,10,11]. Hence, encouraged by the growing awareness of their unique nutritional profile and health benefits, new niche markets have recently emerged, including the segment of hempseed-based food [7].

The cold-pressed hempseed oil successfully fits with the growing consumer interest in those natural cold-pressed oils, which are considered functional foods for the various health benefits provided over their refined counterparts [12,13]. Furthermore, the production of such seed oil turns out to be in line with the uprising trend of the oil industry of seeking out processes that can minimize the environmental impact, decrease toxic residues, and use by-products more efficiently without compromising the nutritional and organoleptic quality of the final product [14,15]. According to the literature, this edible oil exhibits up to 80% polyunsaturated fatty acids (PUFA), mainly linoleic and α-linolenic acids. There are also discrete amounts of γ-linolenic and stearidonic acids, which are only found in certain seed oils [16,17]. Interestingly, the dietary intake of stearidonic acid is beneficial to consumer health as it promotes the synthesis of eicosapentaenoic acid more effectively than α-linolenic acid [18]. Minor bioactive compounds retained during the cold pressing have also been highlighted. These include fat-soluble vitamins, phytosterols, polyphenols, pigments, and inorganic elements. Depending on the type of compound, they may be beneficial to the consumer’s health [19], improve the antioxidative status of the oil [20,21,22], or accelerate its oxidation and quality deterioration [23].

Indeed, owing to the low degree of processing of the raw material, cold-pressed oils contain antioxidants (i.e., tocopherols, polyphenols, and squalene) effective against the lipid oxidation caused by the high PUFA content [19,24], as well as pro-oxidant components (i.e., metals, chlorophylls, and lipid peroxides) responsible for a lower oxidative stability of such oils compared to their refined counterparts [25]. On this basis, cold-pressed hempseed oil tends to have an even higher initial oxidation state than other cold-pressed seed oils (e.g., rapeseed and sunflower), due to a very high unsaturation degree and a large amount of photosensitizer chlorophylls, making it more susceptible to the formation of off-flavors and substances harmful to consumer health [17,24]. For these reasons, cold-pressed hempseed oil necessitates storage in dark or opaque containers. However, to enhance its appeal and consumer acceptance, it is generally marketed in transparent bottles, which inevitably lead not only to an unpleasant color change but also to a faster oxidative deterioration and rancid flavor [17,26].

A few recent attempts focused on ameliorating the oxidative stability of the cold-pressed hempseed oil, and they included the addition of natural antioxidants to the oil [27,28], the optimization of the storage conditions [29], and the improvement of the oil processing [23,26,30]. Regarding the last aspect, ultrasound bleaching has proven to be convenient because it prevents the lipid oxidation catalyzed by pigments by removing the chlorophylls typically abundant in hempseed oil. However, according to the Codex Alimentarius Standard 210-1999 and Amendments (2003/2005), the cold-pressed oil should be obtained only by mechanical processes, such as squeezing or pressing, without the application of heat, and it should be purified only by washing with water, settling, filtering, or centrifuging. Therefore, as an alternative, sustainable, and quick approach to pigment removal, the cold-pressed hempseed oil may go through filtration.

To the best knowledge of the authors, the effect of filtration on the compositional characteristics and the oxidative status of cold-pressed oils has been unfairly underexplored [31]. Hence, the aim of this study was to comprehensively evaluate the effectiveness of filtration on the oxidative stability and minor compounds of the cold-pressed hempseed oil during a 12-week storage in transparent glass bottles. To this purpose, the hydrolytic and oxidative status, fatty acid (FA) composition, tocopherols, pigments, phenols, squalene, and inorganic elements were monitored in experimental filtered and non-filtered oils over the study period, statistically elaborated, and discussed to evaluate the effectiveness of the filtration process on the cold-pressed hemp oil as well as its convenience for future commercial applications.

## 2. Materials and Methods

### 2.1. Production of Non-Filtered and Filtered Cold-Pressed Hempseed Oils

Virgin cold-pressed hempseed oils were provided in 2022 by Sativa Molise (Palata, Italy), and they were obtained from seeds of *C. sativa* L. subsp. Sativa cv. Finola. The seeds were mechanically separated from inflorescences (~95% pure seed and ~5% dockage), dried at 25 ± 2 °C up to a humidity of ~8–10%, packaged in polyethylene bags, and stored in a dry room (21 ± 3 °C, 20–30% RH) for a maximum of two weeks. Hempseeds were cold-pressed in an expeller press with a 1.8 kW electric motor and a capacity of 5–7 kg/h. The press was first heated to 80 °C, then the heater was turned off and the material to be pressed was released. At that time, the temperature of the press head was 40 °C, while the temperature of the outflowing oil was ~50 °C, as an effect of pressure and friction. The speed of the screw press was operated at a constant rotating speed of 25 rpm. The obtained oil was then decanted into a storage tank for 24 h. To produce filtered oil samples, the cold-pressed oil was filtered within 24 h of pressing through a filter press equipped with cellulose acetate membranes (thickness: 0.81 mm) and operating at a pressure of 6 bar.

### 2.2. Samples

Non-filtered and filtered hempseed oils (respectively, NF-HO and F-HO) were separately blotted in triplicate by employing transparent glass bottles of 100 mL each with screw caps (~3% headspace), for a total of *n* = 12 NF-HO bottles and *n* = 12 F-HO bottles. Under these experimental conditions, the oil oxidation may be attributable to autoxidation—occurring just with atmospheric oxygen (^3^O_2_)—and photooxidation—related to the presence of light, sensitizers, and ^3^O_2_.

Changes in the oxidative state and minor compounds of various oil samples were monitored during 12 weeks of storage by keeping all bottles in a controlled environment at room temperature (22 ± 1.2 °C), under a 12/12 h light/dark regime, and rotating them every 10 days [29,32]. Hence, three bottles from every treatment were considered at the beginning of the experimental trial (T0) and every 4 weeks (T4, T8, and 12), and the oil from each bottle was analyzed in triplicate.

### 2.3. Materials and Reagents

Solvents with reagent grade (i.e., n-heptane, n-hexane, diethyl ether, and methanol) were purchased from J.T. Baker (Phillipsburg, NJ, USA), while solvents with HPLC grade (i.e., n-hexane, ethyl acetate, methanol, and water) were supplied by LiChrosolv (Merk, Darmstadt, Germany). Reagents with a trace metal analysis grade [i.e., H_2_O_2_ (30% *v*/*v*) and HNO_3_ (65% *v*/*v*)] and ultrapure water (resistivity of 10 mΩ cm) were purchased from J.T. Baker (Milan, Italy). The Folin–Ciocalteu reagent was from Sigma-Aldrich (Steinheim, Germany). Fatty acid methyl esters (FAMEs) reference standards (C4–C24) and commercial standards of single tocopherols (i.e., α-tocopherol, γ-tocopherol, and δ-tocopherol, 98% purity each), gallic acid (≥99% purity), and squalene (≥98% purity) were purchased from Supelco (Bellefonte, PA, USA). Stock solutions of Na, Mg, K, Fe, Cu, Mn, Zn, Se, Ni, Cr, Al, As, Cd, and Pb (1000 mg/L in 2% HNO_3_, each) were provided by Fluka (Milan, Italy). Depending on the targeted analyte, the internal standards employed for the normalization of the calibration procedure were: tetradecane (99% purity, Sigma-Aldrich) and rhenium (1000 µg/mL in 5% HNO_3_, LGC Standards).

### 2.4. Physicochemical Properties

For the determination of free acidity and peroxide value (PV), the procedures already reported by Costa and colleagues [33] were followed. For the acidity, 90 mL of a solution of ethyl alcohol/diethyl ether (1:2, *v*/*v*) was mixed with a few drops of 1% phenolphthalein and subsequently neutralized with 0.1 N KOH. The mixture was then added to 5 g of the oil sample and titrated with 0.1 N KOH until the color changed. The acidity was calculated according to the following equation and expressed as a % of oleic acid:$$Oleic\;acid\;(\%)=\frac{V\times N\times {MW}_{OA}}{W_{S}\times 10}$$
where **V** is the volume of titrant (mL of KOH), **N** is the normality of KOH (0.1), **MW_OA_** is the molecular weight of oleic acid (282 g/mol), and **Ws** is the weight of the oil sample (g).

For the determination of PV, 25 mL of a solution of glacial acetic acid/chloroform (3:2, *v*/*v*) was mixed with 500 μL of a saturated KI solution. After vigorously shaking, the solution was allowed to stand in the dark for ~5 min. Then, 75 mL of distilled water and starch indicator were added to the mixture, and a titration with 0.01 N Na_2_S_2_O_3_ was conducted until the color changed. PV was defined as milliequivalents of reactive oxygen content per 1 kg of oil sample (mEq/O_2_/kg), and it was derived by following the equation:$$PV\;(mEq/O_{2}/Kg)=\frac{V\times N\times 1000}{W_{S}}$$
where **V** is the volume of titrant (mL of Na_2_S_2_O_3_), **N** is the normality of Na_2_S_2_O_3_ (0.01), and **Ws** is the weight of the oil sample (g). 

Finally, the spectrophotometric exam was conducted by measuring specific UV absorbances at 232 and 270 nm and expressing them as extinction coefficients K232 and K270.

### 2.5. FA Composition

For the elucidation of the FA profile of hemp oil samples, the protocol already employed by Sdiri and colleagues [34] was considered. Approximately 0.1 g of hemp oil was mixed with 2 mL of n-heptane and 0.2 mL of methanolic KOH solution for 30 s at room temperature and decanted. Then, the upper layer containing FAMEs was injected into a gas chromatograph (GC) (Dani Master GC1000) equipped with a split/splitless injector and a flame ionization detector (FID) (Dani Instrument, Milan, Italy). For the chromatographic separation, a SLB-IL100 capillary column (60 m × 0.25 mm ID, 0.20 μm film thickness, Supelco, Sigma Aldrich, Burlington, MA, USA) was employed with the following operating conditions: column oven temperature from 165 °C to 210 °C at 2 °C/min (10 min hold); injector and detector temperatures of 250 °C; He gas at a linear velocity of 30 cm/s (constant); injection volume of 1 μL, with a split ratio of 1:100. Data acquisition and handling were performed using Clarity Chromatography Software v4.0.2. FAMEs of nutritional interest were identified by direct comparison with the retention times of reference compounds and expressed as the relative percent area of the total chromatogram. 

### 2.6. Tocopherols

For the determination of α-, γ-, and δ-tocopherols, ~200 mg of oil sample was diluted in 1.8 mL of *n*-hexane, filtered through a 0.20 μm PTFE syringe filter, and analyzed by a high-performance liquid chromatography system coupled to fluorescence detector (HPLC-FD, Shimadzu, Milan, Italy) according to the conditions already reported by Albergamo and coworkers [35]. Specifically, the chromatographic separation was carried out by a LiChrosorb^®^ Si60 column (250 mm × 4.6 mm I.D., 5 µm particle size, Merck, Darmstadt, Germany), protected by a LiChroCART 4-4 guard column with the same stationary phase (Merck, Darmstadt, Germany), and by exploiting a mobile phase consisting of n-hexane/ethyl acetate (90:10 *v*/*v*) under isocratic conditions. HPLC-FD analyses were performed at 40 °C with an injection volume of 20 µL and a flow rate of 0.8 mL/min. Data processing occurred by the LabSolutions software, ver. 5.10.153 (Shimadzu). The identification of tocopherols was carried out by a direct comparison with the retention time of relative commercial standards at respective excitation and emission wavelengths of 295 nm and 330 nm. The quantitative analysis was performed by constructing appropriate external calibration curves for every investigated tocopherol.

### 2.7. Pigments and Polyphenols

For the determination of chlorophyll (Chl) a, chlorophyll (Chl) b, and total carotene, the protocol of Blasi and colleagues [19] was considered. Briefly, 1 g of every oil sample was mixed with 50 mL of diethyl ether, vortexed, and sonicated for 1 min. The absorbance of solutions was measured by an UV spectrophotometer (UV-2401 PC, Shimadzu, Milan, Italy). **Chl a** and **Chl b** showed the maximum absorbances at 663 nm (A663) and 640 nm (A640), respectively, while total carotene content was determined at 470 nm (A470). The concentration (μg/mL) of these pigments was calculated according to the formulas proposed by Izzo et al. [20]: Chl a=9.93×A663−0.78×A640
Chl b=17.60×A640−2.81×A663
Chl a+b=7.12×A663−16.80×A640
Total carotene=(1000×A470−0.52×Chl a−7.25×Chl b)/226

For the extraction of polyphenols, approximately 6 g of oil samples were mixed with 6 mL of a methanol/water solution (80:20, *v*/*v*), stirred for 2 min, and kept at room temperature until phase separation. Then, the colorimetric assay was conducted according to what had already been reported by Aghraz and coworkers [36]. Specifically, 0.2 mL of the upper part of the mixture was collected, and 1.8 mL of distilled water was added along with 8 mL of Na_2_CO_3_ (20%) and 10 mL of Folin–Ciocalteu reagent. The mixture was kept in the dark for 30 min and read at 700 nm with an UV-visible spectrophotometer (UV-2401 PC, Shimadzu, Milan, Italy). The quantification procedure occurred through an external calibration curve of gallic acid, and the total phenol content was calculated as milligrams of gallic acid equivalent in 1 L of hemp oil (mg GAE/kg).

### 2.8. Squalene

Squalene was extracted from every oil sample by means of a solid phase extraction (SPE) exploiting Supelco Discovery DSC-Si Silica cartridges and n-hexane and analyzed by a gas chromatography system (GC-2010, Shimadzu, Tokyo, Japan) coupled to a single quadrupole mass spectrometer (QP-2010 Plus, Shimadzu, Japan) according to the protocol reported in Vadalà and colleagues [37]. Chromatographic separations occurred on a SPB-5 MS capillary column (30 m × 0.25 mm i.d. × 0.25 μm film thickness, Supelco, Bellefonte, PA, USA). The oven temperature program was from 80 °C (1 min hold) to 140 °C at 20 °C/min, and finally to 290 °C (2 min hold) at 5 °C/min. The injection port temperature was set at 250 °C, and the injection volume was 1 μL with a split ratio of 1:10. The MS conditions were: EI source temperature 230 °C; ionization energy and emission current 70 eV and 250 μA, respectively; interface temperature 290 °C. The identification occurred in full scan (mass range: 40–400 *m*/*z*) by comparing both retention time and mass spectrum with those of commercial standards, while quantification was performed in selected ion monitoring (SIM) by monitoring four characteristic mass fragments (121, 137, 161, and 175 *m/z*). Hence, the amount of the compound was derived by considering the relative base peak ion and exploiting the internal standard normalization with the internal standard tetradecane.

### 2.9. Element Analysis

Around 0.5 g of each oil sample were mineralized with 8 mL of HNO_3_ and 2 mL of H_2_O_2_ by a microwave digestion system (Ethos 1, Milestone, Bergamo, Italy) with a temperature of 0–200 °C in 10 min (step 1) and 200 °C held for 10 min (step 2) and a power of 1000 W. Digested samples were cooled down at room temperature and properly diluted with the internal standard Re in ultrapure water. Elemental analyses were carried out by a quadrupole ICP-MS iCAP Q (Thermo Scientific, Waltham, MA, USA), equipped with an ASX-520 autosampler (Cetac Technologies Inc., Omaha, NE, USA). Samples were screened in triplicate for selected elements according to the procedure reported in previous works [38,39] by using the following operating parameters: incident radio frequency power equal to 1500 Wand plasma, auxiliary gases, and carrier gases (Ar) at respective flow rates of 15 L/min, 0.9 L/min, and 1.10 L/min. The instrument operated in He collision mode (4 mL/min) and with a spray chamber set at +2 °C. The injection volume and the sample introduction flow rate were, respectively, 200 µL and 1 mL/min. Spectra acquisition was performed in full scan mode (dwell time 0.5 or 0.01 s/point, depending on the analyte). For quantification purposes, an external calibration procedure combined with an internal standard normalization was exploited. Instrumental control and data acquisition were performed by Thermo Scientific’s Qtegra™ Intelligent Scientific Data System software (Thermo Fisher Scientific, Bremen, Germany).

### 2.10. Statistical Analysis

The data were statistically analyzed by R Studio v. 3.6.1 (Boston, MA, USA) for Windows. A descriptive analysis, including the mean and standard deviation, was conducted for all the experimental data obtained from this study. After running a Shapiro–Wilk test to verify the normal distribution of experimental data, every parameter was statistically elaborated in all oil samples by (i) the one-way ANOVA followed by a post hoc Tukey’s HSD to study the effect of storage and highlight significant differences in NF-HO samples (or F-HO samples) during T0-T12, and (ii) a two-tailed Student’s *t*-test for unpaired data to evaluate the effectiveness of filtration and point out significant differences between NF-HO and F-HO samples. Statistical significance was accepted at *p* ≤ 0.05.

## 3. Results

### 3.1. Hydrolytic and Oxidative Status of Cold-Pressed Hempseed Oils

The hydrolytic and oxidative status of cold-pressed hempseed oil was explored in NF-HO and F-HO samples (Figure 1) and compared with the available quality parameters fixed for edible fats and oils not covered by individual standards by the Codex Alimentarius for free acidity (2% of oleic acid or 4 mg KOH/g of oil) and PV (15 mEqO_2_/kg of oil) [40].

Free fatty acids are more susceptible to autooxidation than esterified fatty acids, and they also stimulate the hydrolysis of phenolics [25,41], thus contributing to the deterioration of the shelf life of the edible oil. At the beginning of the experimental trial, fresh NF-HO and F-HO samples were characterized by similar and acceptable acidities (respectively 0.87% and 0.90%). By exploiting the highest formation rate during the first 4 weeks of storage (NF-HO: +54%; F-HO: +49%), free fatty acids increased, but still within the Codex limits, in both types of oil (NF-HO: 0.87–1.87%, *p* < 0.05; F-HO: 0.90–1.77%, *p* < 0.05), in agreement with the recent literature, which, however, has also reported fresh oils with a very high acidity [20,29,42,43]. With the advancement of storage (T4-T12), lower formation rates were recorded, and only NF-HO reported acidities higher than the Codex guide value, the free acidity of NF-HO and F-HO samples being respectively from 1.87% to 4.30% (*p* < 0.05) and from 1.77% to 1.93% (*p* < 0.05) (Figure 1a,b). Overall, the NF-HO showed a worse hydrolytic quality than the F-HO (mean acidities: 2.52% and 1.60%, *p* < 0.05), thus suggesting an ameliorative effect of filtration on the oil acidity (Figure 1c).

Notoriously, proper seed storage conditions (in terms of humidity, light and temperature) as well as good seed quality (in terms of moisture level and percent dockage) are essential requirements to produce an oil with a favorable hydrolytic status [44]. However, according to the results of this study, the filtration also had a positive effect on the acidity of the cold-pressed hempseed oil during storage, probably because less water was dispersed in the filtered oil samples. This would be consistent with the recent work of Tura and colleagues [29], in which stable and compliant acidities were recorded in cold-pressed hempseed oils subjected in a laboratory to cotton gauze filtration and centrifugation before storage in amber glass bottles for 90 days. Additionally, Fregapane et al. [45] reported that filtration reduced the rate of hydrolysis of the triacylglycerol matrix with positive effects on the oxidative stability of the virgin olive oil. However, on the other hand, Frega and colleagues [46] suggested that filtration increased the oil’s susceptibility to oxidative degradation by removing suspended solid materials. From their point of view, suspended solids and free fatty acids may react to form a precipitated residue that is not capable of an oxidative reaction.

PV is indicative of the total peroxidic-bonded oxygen present in an oil and, consequently, of its oxidative status, especially at the early stages of storage. As a result, lipid peroxyl radicals and hydroperoxides are typically defined as primary oxidation products. This parameter can vary in relation to the oxygen partial pressure in the headspace, the type of oxygen (i.e., ^1^O_2_ is much more reactive with lipids than ^3^O_2_), temperature (i.e., the solubility of oxygen in oil generally increases as the temperature increases [47]), light (i.e., light promotes production of ^1^O_2_ in the presence of sensitizer pigments and ^3^O_2_), and minor oil components (i.e., metals, free fatty acids, mono- and diacylglycerols, and phospholipids generally accelerate oil oxidation). Fresh NF-HO and F-HO samples had similar and acceptable PVs (respectively 7.98 mEqO_2_/kg and 8.21 mEqO_2_/kg). However, similarly to the acidity, they showed an increase within 4 weeks of storage in both sets of samples (NF–HO: 7.98–14.39 mEqO_2_/kg, *p* < 0.05, and F–HO: 8.21–12.64 mEqO_2_/kg, *p* < 0.05), while remaining lower than the Codex value. Values from fresh hempseed oils can be hardly compared with literature since peroxides quickly react to generate other radical forms, thus showing a considerable fluctuation even in the same oil [20,29,32,42,48]. Interestingly, during the first 4 weeks of storage, the highest rate of peroxide formation was observed both in NF-HO (+45%) and F-HO (+35%) samples. This can be interpreted as a trigger effect generated by the initial and variable level of ^3^O_2_ dissolved in the oil during the production and bottling phases [49]. With the advancement of storage (T4-T12), however, the PV of NF-HO samples became higher than the Codex guidance value. Indeed, PV of NF-HO and F-HO samples varied respectively from 12.82 mEqO_2_/kg to 16.85 mEqO_2_/kg (*p* < 0.05) and from 7.64 mEqO_2_/kg to 13.38 mEq O_2_/kg (*p* > 0.05) (Figure 1a,b). Overall, the effectiveness of filtration on oil peroxides is also supported by the fact that F-HO showed a better oxidative status than NF-HO (mean PVs: 12.18 mEqO_2_/kg and 15.49 mEqO_2_/kg, *p* < 0.05) (Figure 1c).

Specific extinction coefficients (K) at the UV wavelengths of 232 nm and 270 nm are helpful in studying the progress of the autooxidation of vegetable oils. In fact, when hydroperoxides are formed, double bond shifting and isomerization occur, producing primary oxidation products such as conjugated dienes, which exhibit intense absorption at 232–234 nm, and subsequently conjugated trienes, which typically adsorb at 268–270 nm [50]. At this point, the decomposition of conjugated systems into secondary oxidation products (i.e., aldehydes, ketones, alcohols, and short-chain hydrocarbons) is commonly observed. According to our results, the extinction K_232_ rose from 1.88 to 4.10 (*p* < 0.05) in NF-HO and from 2.41 to 2.87 (*p* < 0.05) in F-HO. Similarly, K_270_ of NF-HO and F-HO samples increased, respectively, in the ranges 0.74–0.81 (*p* < 0.05) and 0.53–0.71 (*p* < 0.05). In line with previous studies on hempseed oil, both NF-HO and F-HO were characterized by K_232_ and directly correlated with the evolution of PV [23,29]. Moreover, similarly to PV, K_232_ and K_270_ showed the highest increase during the first 4 weeks of storage, both in NF-HO (+43% and +36%) and F-HO (+32% and +30%) samples. Subsequently, a steady state of oxidation was observed during storage. In fact, K_232_ and K_270_ were not significantly different both in NF-HO and F-HO during T4-T8 (*p* > 0.05), probably due to the establishment of an equilibrium between the formation of diene systems and their decomposition into secondary products, such as hexanal, 2-decenal, or 2-heptenal, potentially responsible for the off-flavor of the oxidized oils [51]. From the discussed data, a weak effect of filtration on the formation of dienes and trienes in the hempseed oil during storage may be argued. This is also suggested by the fact that NF-HO had slightly higher conjugated dienes (mean values: K_232_ = 3.25 vs. 3.12, *p* > 0.05) and trienes (mean values: K_270_ = 0.74 vs. 0.70, *p* > 0.05) than F-HO (Figure 1c). A possible explanation of the different oxidative behavior of hempseed oils is that filtration may reduce all those minor oil components that may cause the initial increase in PV and conjugated systems by various mechanisms, such as autooxidation (i.e., free fatty acids), increase of the rate of oil oxidation (i.e., transition metals), and increase of the diffusion rate of oxygen from the headspace into the oil (i.e., free fatty acids, mono- and diacylglycerols, and phospholipids) [25]. Consequently, a greater presence of these compounds in NF-HO samples would be indicative of lower stability against oxidative degradation. Additionally, NF-HO would be more affected by the light from the use of transparent bottles than F-HO, since light may trigger _1_O^2^ oxidation pathways and, consequently, further increase the level of primary oxidation products [25,52].

Conflicting conclusions on the influence of filtration and storage on the oxidative stability of the oil can be drawn from the literature. A study from Shendi and coworkers [53] was consistent with our findings since PV, K_232_, and K_268_ significantly differed in filtered and non-filtered olive oils during two years of storage in amber glass bottles, with the highest values observed in non-filtered oil samples at the early months of storage. Liang and colleagues [23] did not employ filtration but ultrasonic bleaching to reduce the amount of prooxidant components in the hempseed oil, and they found that such a treatment had an impact on the oxidative oil stability as it significantly delayed the increase both in PV and conjugated diene systems, compared to untreated control oils during accelerated storage in colored glass containers. However, filtration also showed to have no effect on oil peroxides, dienes, and trienes [45,54,55,56] or even accelerate the formation rate of such compounds [56,57,58] in olive oils.

### 3.2. FA Composition

Table 1 reports the profile of the most nutritionally relevant FAs present in NF-HO and F-HO samples stored for 12 weeks in transparent glass bottles.

Minimal differences were recorded in single FAs of NF-HO and F-HO samples at T0. However, major classes of SFA and MUFA were more concentrated in NF-HO than F-HO (SFA: 10.44% and 9.72%, MUFA: 11.35% and 10.18%); conversely, PUFA were higher in F-HO than NF-HO (81.87% and 80.74%). During storage of NF-HO samples, no significant variations were detected between the initial (T0) and final (T4) percent content of most single FAs. In fact, only the palmitic and linoleic acids decreased significantly over 12 weeks of storage (i.e., 7.20–5.87% and 55.56–52.36%, *p* < 0.05) and, in parallel, total SFA and PUFA reduced significantly from 10.44% to 8.27% (*p* < 0.05) and from 80.74% to 77.71% (*p* < 0.05). Accordingly, the n-6/n-3 ratio was lowered from 3.07 to 2.07 (Table 1). Conversely, in F-HO samples, while most FAs decreased, the linoleic acid remained stable (56.27–54.29%, *p* > 0.05) and the stearidonic acid increased (1.02–1.76, *p* < 0.05) during the study period. Accordingly, SFA, MUFA, PUFA, and n-6/n-3 significantly decreased in the respective ranges of 9.72–7.80% (*p* < 0.05), 10.18–9.29% (*p* < 0.05), 81.87–80.06% (*p* < 0.05), and 3.05–2.84 (*p* < 0.05).

As shown in Figure 2, the filtration process impacted the FA composition. In fact, SFA and MUFA were on average more abundant in NF-HO than F-HO (respectively, 9.24% and 8.37%, *p* < 0.05; 10.45% and 9.67%, *p* < 0.05). Conversely, PUFA and PUFA/SFA were lower in NF-HO and higher in F-HO (respectively, 78.97% and 80.50%, *p* < 0.05; 8.61% vs. 9.64%, *p* < 0.05).

Overall, the FA composition of the cold-pressed hempseed oil, revealed in both NF-HO and F-HO samples, was within the ranges reported by recent literature, which already highlighted high levels of linoleic (50–80%) and α-linolenic (15–25%) acids in the hempseed oil [8,59,60], as well as a peculiar amount of stearidonic acid (0.5–1.5%) [60,61]. In addition, the oil from this study belonged to the Finola hemp variety, which has already been shown to contain up to 2% stearidonic acid and up to 4% γ-linolenic acid [62,63,64].

Vegetable oils with a high unsaturation degree are characterized by a greater formation rate and amount of primary oxidation compounds accumulated over time. The rates of autooxidation and _1_O^2^ oxidation depend on the rate of the lipid alkyl radical formation, which in turn depends mainly on the type of FA [25]. In this respect, unsaturated FAs, such as oleic, linoleic, and linolenic acids, are particularly prone to oxidation and consequently more easily degraded in oil [64,65]. This would explain the reduction of MUFA and PUFA observed in the hempseed oils during 12 weeks of storage and, in general, the lower oxidative stability of high PUFA oils [32]. The factor “light” must also be considered during storage since it is well known that the oxidation of unsaturated FAs is accelerated by exposure to light, especially when pigments, such as chlorophylls, are present in the oil [66]. The influence of light on the FA composition of hempseed oil has not been investigated in any study. However, Rastrelli and colleagues [66] found out that the sum of linoleic and α-linolenic acids decreased more significantly in the clear glass oil bottles than in the dark ones during 1 year of storage. Therefore, under our experimental conditions, both the oxygen dissolved in the oil and the light may contribute to the decreasing stability of unsaturated FAs in the hempseed oil.

Considering the filtration process, no significant differences in single FAs were revealed between F-HO and NF-HO, thus confirming that oil treatments do not cause drastic changes in the FA profile of a vegetable oil [67,68,69]. However, Golimowsky and colleagues [62] recently applied a low-temperature bleaching to the cold-pressed hempseed oils with positive effects on the major classes of FAs rather than single FAs, as the oils displayed a reduced SFA proportion and a consequent growth in the proportion of PUFA and SFA/PUFA ratio. Accordingly, results from this study pointed out that filtration may better preserve the MUFA and PUFA of the cold-pressed hempseed oil during storage.

### 3.3. Tocopherols

Tocopherols are the most important antioxidants present in vegetable oils since they reduce the extent of oil autooxidation by competing with unsaturated FAs for alkoxyl and peroxyl radicals in synergistic action with polyphenols [70]. In the present study, the content of α-, γ-, and δ-tocopherol in NF-HO and F-HO samples stored for 12 weeks in transparent glass bottles is illustrated in Figure 3. As expected, fresh NF-HO and F-HO displayed very different contents of α- (42.03 mg/kg and 8.44 mg/kg), γ- (1059.56 mg/kg and 172.94 mg/kg), and δ- (33.37 mg/kg and 6.44 mg/kg) tocopherols. Additionally, a further reduction in all isomers occurred in all oils with increasing storage time, especially at the initial stage (T0-T4). Specifically, γ-tocopherol decreased in NF-HO from 1059.56 mg/kg to 549.16 mg/kg (*p* < 0.05) over the study period, with a 24% reduction observed during T0-T4, while in F-HO it showed a reduction from 172.94 mg/kg to 110.14 mg/kg (*p* < 0.05), with a 12% loss observed during the first 4 weeks of storage. Alpha-tocopherol varied in NF-HO from 42.03 mg/kg to 14.64 mg/kg (*p* < 0.05) during 12 weeks of storage, lowering by 28% during T0-T4. The same isomer was reduced in F-HO from 8.44 mg/kg to 4.25 mg/kg (*p* < 0.05), with an 18% decrease during T0-T4. Delta-tocopherol was lowered in the range 33.37–24.31 mg/kg (*p* < 0.05) and 6.44–3.76 mg/kg (*p* > 0.05), respectively, in NF-HO and F-HO samples stored over T0-T12, with respective reduction rates of 19% and 9% during the first 4 weeks of storage (Figure 3a,b). Differently from other isomers, α-tocopherol showed higher degradation rates in NF-HO along with statistically significant quantitative changes (*p* < 0.05) observed at every storage step. Conversely, it had a lower consumption rate in F-HO samples, together with relatively stable contents during T4-T12 (Figure 3a,b). Overall, the filtration procedure demonstrated that it markedly affected the content of tocopherols in oil. In fact, NF-HO samples showed higher mean tocopherol contents than F-HO (γ-tocopherol: 790.00 mg/kg vs. 139.61 mg/kg, *p* < 0.05; α-tocopherol: 27.82 mg/kg vs. 6.21 mg/kg, *p* < 0.05; δ-tocopherol: 27.90 mg/kg vs. 5.47 mg/kg, *p* < 0.05) (Figure 3c). Additionally, from the data discussed above, it is evident that filtration affected the degradation rate of tocopherols, since it was generally more pronounced in NF-HO than F-HO samples. Cold-pressed hempseed oils from this study shared the highest content of γ-tocopherol followed by α- and δ- isomers, thus resulting in line with the previous literature [19,20,29]. However, while the freshly produced NF-HO samples were marked by tocopherol levels consistent with most recent studies dealing with non-treated hempseed oils (Table 2), the F-HO samples were characterized by much lower amounts of such antioxidants. However, differently from this study, Tura and colleagues [42] did not find significant differences between filtered and non-filtered commercial hempseed oils.

Considering the evolution of tocopherols during storage, our findings are consistent with previous literature on vegetable oils, including hempseed oil. Tura and colleagues [29] recorded a non-statistically significant reduction of γ- and α-tocopherols and a statistically significant reduction of the δ-isomer in cold-pressed hempseed oils subjected in a laboratory to filtration and centrifugation before storage in amber glass bottles for three months. In another work, however, Tura and coworkers [48] demonstrated that α-tocopherol was present in the oil only before starting an accelerated oxidation test, while γ-tocopherol significantly decreased, with the main decrease noticed during the first 6 days of the test (corresponding to 6 months of storage at room temperature). Interestingly, Rastrelli and colleagues [66] assessed the change in minor compounds of extra-virgin olive oil stored at different conditions and for 12 months, and they pointed out that α-tocopherol decreased by 92–93% in half-empty clear bottles and 24–25% in filled clear bottles, thus confirming that the contribution of light to the direct decomposition of such tocopherol was negligible.

The decrease in tocopherols observed in NF-HO and F-HO over storage time may be attributed to the role of these antioxidants in counteracting the process of lipid autooxidation [25]. Additionally, under the experimental conditions of the study, their degradation may be accelerated by their protective action against photooxidation processes induced by sensitizer pigments in the presence of light [73]. Indeed, tocopherols cannot be excited by visible light, but they can scavenge _1_O^2^ by combining physical and chemical quenching [74]. Particularly, the tocopherol oxidation rate would vary according to the isomer type, following the order α > γ > δ [75,76,77]. This was also confirmed by our study, since δ-tocopherol was consumed at a slower rate than γ- and α-isoforms in both NF-HO and F-HO, thus making it more stable against oxidation. However, as previously mentioned, greater tocopherol deterioration rates were recorded in NF-HO than in F-HO. This may be because high concentrations of tocopherols are capable of being consumed via their radical forms in side reactions, which may result in a prooxidant effect. In fact, the tocopheroxy radical, especially the α-tocopheroxy radical, subtracts hydrogen from the lipid matrix and produces tocopherol and lipid alkyl radicals, which would accelerate the lipid oxidation [78]. It has also been suggested that such an effect may be promoted by a synergistic action with high concentrations of prooxidant transition metals [79]. From this point of view, filtration would be beneficial as it would prevent oxidation processes mediated by high levels of tocopherols. However, contrary to our findings, Fregapane and coworkers [45] observed similar degradation rates of α-tocopherol in non-filtered and filtered extra-virgin olive oils stored under accelerated conditions for 8 months, thus concluding that filtration had no effect on its degradation over time.

### 3.4. Chlorophylls, Carotenes, and Polyphenols

Figure 4 illustrates the trend of total chlorophylls (intended as the sum of chlorophylls a + b) and total carotenes recorded in NF-HO and F-HO samples over 12 weeks of storage in transparent glass bottles.

The filtration induced a consistent pigment decrease in fresh F-HO compared to the NF-HO counterpart (chlorophylls: 65.44 mg/kg and 18.93 mg/kg; carotenes: 52.70 mg/kg and 37.41 mg/kg). A further reduction was observed with increasing storage weeks. Specifically, total chlorophylls and carotenoids reduced from 65.44 mg/kg to 9.23 mg/kg (*p* < 0.05) and from 52.70 mg/kg to 34.98 mg/kg (*p* < 0.05) in NF-HO samples, with the greatest reduction rates at T0-T4 (respectively, −59% and −17%). A downward trend was also observed in the F-HO samples, as chlorophylls a + b and carotenoids decreased in the ranges of 18.93–7.23 mg/kg (*p* < 0.05) and 37.41–28.15 mg/kg (*p* < 0.05), with the highest deterioration rates equal to −42% and −6% observed after 4 weeks of storage. For both types of pigment, high degradation rates were recorded in NF-HO samples along with statistically significant quantitative changes (*p* < 0.05) during T4-T12. Conversely, F-HO samples displayed lower consumption rates together with relatively stable pigment contents after 4 weeks of storage (Figure 4a,b). Additionally, although a proper color assessment could not be made, fresh NF-HO samples were characterized by a deep and cloudy green color that turned brown at the end of storage, whereas fresh F-HO samples had a more stable color that varied from a brilliant yellow to a slightly darker hue after 12 weeks of storage. As expected, filtration had a great impact on such pigments, as all NF-HO and F-HO samples showed a mean chlorophyll level amounting to 29.13 mg/kg and 11.34 mg/kg (*p* < 0.05) and a mean carotene content equal to 44.18 mg/kg and 32.93 mg/kg (*p* < 0.05) (Figure 4c).

Freshly produced NF-HO samples were marked by a chlorophyll content consistent with non-treated hempseed oils characterized in other recent studies (Table 3). On the other hand, fresh F-HO samples had slightly higher pigment contents than hempseed oils treated by ultrasound bleaching or refining processes (Table 3), thus suggesting that the filtration has a milder effect on such compounds. However, the recent literature showed variable and lower levels of carotenoids with respect to our study, whether or not the hemp seed oil had received any treatment (Table 3). Only Aladić and colleagues [80] obtained similar carotenoid levels (31.5 mg/kg) by producing the cold-pressed hempseed oil on a laboratory scale.

To the best knowledge of the authors, the evolution of pigments in the cold-pressed hempseed oil during storage has not yet been investigated. However, our findings are consistent with the general degradation of these pigments observed during the storage of other vegetable oils [25,81].

Total chlorophylls tend to degrade to pheophytins and pheophorbides, with greater rates observed during the initial stages of storage and upon exposure to light [81,82]. Chlorophylls and derivatives are primarily involved in the photooxidation process since they can act as sensitizers in the presence of light and ^3^O_2_ to produce ^1^O_2_, thus causing a faster deterioration of the oxidative stability, color change, and an inevitable reduction in the nutritional and economic value of the oil [25,81]. On the other hand, carotenoids slow down photooxidation by light filtering, ^1^O_2_ quenching, sensitizer inactivation, and free radical scavenging [25]. Among carotenoids, *β*-carotene could be protected from degradation by *α*-tocopherol, to prevent both the autooxidation and the photooxidation of an oil with a synergistic effect [25,83]. In the presence of chlorophyll, carotenoids counteract the progress of photooxidation by physical or chemical ^1^O_2_ quenching. However, while the physical quenching does not alter their structure, the chemical quenching leads to their degradation and conversion into epoxide or carbonyl derivatives. As a result, an overall reduction in carotenoids in the oil, as well as its discoloration, can be expected to occur during storage and in the presence of light [84].

Results from this study also pointed out that filtration lowered the pigment content of the fresh hempseed oil and, at the same time, reduce its consumption during storage. Indeed, greater pigment deterioration rates were recorded in NF-HO samples than in F-HO samples over the study period. This may be explained by the higher content of chlorophyll in non-filtered oils, which, reacting with the atmospheric oxygen (^3^O_2_) dissolved in the oil, triggers the process of photooxidation to a greater extent, especially at the initial stages of storage. As a result, carotenoids were also consumed at higher rates in non-filtered oils to counteract the oxidation process. Therefore, oil filtration has become highly desirable not only to limit the photooxidation process by chlorophylls and derivatives but also to slow down the consumption of antioxidant carotenoids over time. As proof, Liang and colleagues [23] found that PV were significantly inversely correlated with total chlorophylls in non-treated hempseed oils, while no correlation was observed in the oils treated by ultrasound bleaching, which were characterized by lower and more stable chlorophylls during accelerated storage.

Similarly to pigments, fresh NF-HO showed a higher level of total phenols than fresh F-HO samples, and an overall reduction in such secondary metabolites was observed in both series of hempseed oil over storage. Specifically, total polyphenols decreased in the range 82.88–12.97 mg/kg (*p* < 0.05) and 57.55–28.30 mg/kg (*p* < 0.05) respectively in NFHO and F-HO samples (Figure 4a,b), with higher oxidation rates recorded in NF-HO than F-HO samples, especially in the early storage (T0-T4: −71% and −23%, respectively). However, both arrays of oil samples shared a non-statistically significant reduction in total polyphenols at every storage step analyzed during T4-T12 (*p* > 0.05, Figure 4a,b). Interestingly, F-HO was characterized by a non-statistically significant higher total phenol content than NF-HO (respectively, 41.39 mg/kg and 34.63 mg/kg, *p* > 0.05) (Figure 4c).

The activity of polyphenols is notably supported by the presence of tocopherols in the oil. In fact, a clear synergism between polyphenols, acting both as metal chelators and radical scavengers, and tocopherols, acting as radical scavengers, has already been displayed in previous studies dealing with cold-pressed oils with high PUFA levels [70,85]. However, the antioxidant activity of phenolics is strictly related to the compound family and content, which in turn depend on genotype, geographical origin, cultivation practices of the hemp, and extraction and storage conditions of the oil [43,86,87,88].

While considering that total polyphenols by the Folin–Ciocalteu method might be biased by several interfering non-phenolic compounds, such as amino acids, ascorbic acid, reducing sugars, and transition metals [89]. Highly variable quantitative ranges have been found in recent literature on cold-pressed hempseed oils (Table 3). Consequently, the total polyphenol content of fresh oils from our study would be more similar to that reported by Ciano and coworkers and Tura and colleagues [29,68]. On the other hand, Smeriglio and colleagues [88] reported much higher and non-comparable total phenolics (267,500 mg/kg) in cold-pressed hempseed oil obtained from the same cultivar used in this study.

Considering the trend of polyphenols during oil storage, a recent work by Tura et al. [29] has already pointed out the degradation of such antioxidants in the cold-pressed hempseed oil subjected to accelerated storage. In line with our results, polyphenol deterioration was also strongly linked to the progress of oxidation during the storage of virgin olive oil [52,54,90,91], camelina oil [75], and cold-pressed rapeseed oil [92]. Indeed, these studies revealed that the oxidation of phenolic compounds was linearly related to the formation of primary oxidation products over storage, probably because they were increasingly involved in the scavenging of hydroperoxide radicals. The influence of light on the degradation rate of polyphenols was studied in olive oil stored in dark and transparent glass bottles, and non-significant changes were observed in the content of such secondary metabolites over time [66]. Therefore, it can be concluded that, under our experimental conditions, the oxygen dissolved in the oil and the content of primary oxidation products would most affect the stability of polyphenols [90].

The filtration process had a clear effect on the cold-pressed hempseed oil. From the obtained results, it is evident that filtration can lower the water-soluble phenolic content in the fresh hempseed oil and at the same time reduce its deterioration rate during storage. This was probably because filtration improved the oxidative stability of the hempseed oil, as discussed in paragraph 3.1, with relevant implications for reduced consumption and better stability of the polyphenol content during storage. On the other hand, the lower oxidative stability of non-filtered oils resulted in a faster consumption of these antioxidants.

The literature showed contradictory results on the influence of filtration and storage on the polyphenol content of vegetable oils. Some studies confirmed the progressive decrease in polyphenols during the storage of virgin olive oils. However, higher deterioration rates were recorded in filtered oils than in non-filtered oils, which consequently showed higher total phenol contents [53,54]. Conversely, other research highlighted minimal or no differences in total phenols between filtered and non-filtered olive oils during storage [56,57,58].

### 3.5. Squalene

Figure 5 reports the content of squalene in NF-HO and F-HO samples over the considered storage period. As expected, fresh NF-HO showed a higher amount of squalene than fresh F-HO (557.01 mg/kg and 152.22 mg/kg). However, squalene decreased over storage in both NF-HO and F-HO within the respective ranges of 557.01–280.53 mg/kg (*p* < 0.05) and 152.22–92.38 mg/kg (*p* < 0.05), again with more pronounced reduction rates observed in NF-HO than F-HO samples, especially during the first 4 weeks of storage (respectively, −23% and −21%) (Figure 5a,b). On average, NF-HO was characterized by a higher amount of squalene than F-HO (respectively, 401.74 mg/kg and 117.71 mg/kg, *p* < 0.05) (Figure 5b).

Little and recent literature [93,94,95] has addressed the squalene in the hempseed oil, reporting levels of this antioxidant not comparable to those found in this study (Table 4), because of the influence of genetic, environmental, and agronomic factors [96]. Additionally, oil refining, if present, significantly reduced the content of such phytochemicals in vegetable oils [97].

To the best knowledge of the authors, the influence of filtration and storage on the content of squalene has not yet been explored in minor cold-pressed oils. However, the consumption of this antioxidant typically occurs during the storage of virgin olive oil, and the diffuse lighting of the oil did not seem to play a major role in the degradation process [66]. As a result, the consumption of this compound observed in the hempseed oils during storage would be attributable to the scavenging of peroxyl radicals [98] and the consequent conversion into quite stable degradation products that are not involved in further propagation reactions [99]. As proof, the olive oil refined and subsequently enriched with squalene showed lower acidity, PV, and absorbances at 237 nm and 270 nm with respect to the crude olive oil during storage, thus confirming the role of such antioxidants in the delay of autooxidation [100].

### 3.6. Inorganic Elements

The profile of inorganic elements in NF-HO and F-HO samples recorded during storage is reported in Table 5.

Freshly produced NF-HO and F-HO samples were characterized by similar element profiles. Specifically, Na was found at the highest content (NF-HO: 238.95 mg/kg and F-HO: 254.14 mg/kg), followed by other major elements, such as K (NF-HO: 46.40 mg/kg and F-HO: 45.39 mg/kg), and Mg (NF-HO: 13.99 mg/kg and F-HO: 12.92 mg/kg). Among essential trace elements, the most abundant elements were Fe (NF-HO: 4.11 mg/kg and F-HO: 5.28 mg/kg) and Zn (NF-HO: 7.86 mg/kg and F-HO: 7.95 mg/kg). Considering the potentially toxic trace metals, Al was the most concentrated metal (NF-HO: 5.74 mg/kg and F-HO: 4.58 mg/kg), followed by a much lower amount of Pb (NF-HO: 0.26 mg/kg and F-HO: 0.41 mg/kg). Additionally, both fresh NF-HO and F-HO samples were marked by levels of Pb and As lower than the limit fixed by the Codex Alimentarius for such heavy metals (0.1 mg/kg) [40].

For greater convenience, only the data from T0 and T12 were reported in Table 5, as both NF-HO and F-HO samples showed overlapping and non-statistically different element contents (*p* > 0.05) from the start to the end of the storage period. Not only the storage in clear glass bottles but also the filtration process did not alter these minor components in the cold-pressed hempseed oil since, as reported in Figure 6, all NF-HO and F-HO samples showed non-significantly different mean concentrations of each investigated analyte (*p* > 0.05).

To the best knowledge of the authors, the effect of storage and filtration processes on the inorganic elements of the cold-pressed hempseed oil during storage has not been addressed elsewhere, nor is there any comparative literature on other vegetable oils. The stable and comparable element profile obtained during the storage of both NF-HO and F-HO may be related (i) to the absence of inorganic matter decomposition during the storage of the oils and, not least, (ii) to the non-responsiveness of these minor compounds to the filtration process.

However, this compositional aspect should be worthy of more in-depth investigation, as the oxidative stability of edible oils is influenced not only by storage conditions, energy inputs (e.g., light), oxygen type, and triacylglycerol matrix but also by a variety of minor compounds, including metals.

In fact, metals catalyze the initiation step in the autooxidation, and, not least, they produce _1_O^2^ and hydroxy radicals, respectively, from _3_O^2^ and hydrogen peroxides, which inevitably accelerate the overall oil oxidation. Among elements, transition elements such as Fe and Cu are known to accelerate these processes [25]. Additionally, Fe also causes the decomposition of phenolics, thus further deteriorating the oil’s oxidative stability [101].

## 4. Conclusions

The present study showed that the filtration treatment could improve the oxidative stability and extend the shelf life of the hempseed oil bottled in transparent glass, resulting in an effective and sustainable alternative to other severe refining procedures in view of maintaining the good nutritional standards typical of cold-pressed oils. Benefits from the hempseed oil filtration include, certainly, an improvement in the hydrolytic and oxidative status, as well as a better preservation of MUFA and PUFA content within 12 weeks of storage. By removing a large amount of chlorophyll, filtration can not only produce lighter and brighter-colored oils but also promote the use of clear glass bottles for packaging, which has a positive impact on consumer acceptance. As expected, fresh filtered oils were characterized by reduced carotenoids, tocopherols, polyphenols, and squalene. However, findings from this study highlighted the “protective effect” of filtration on such antioxidants, resulting in lower degradation rates and better preservation of these minor compounds than crude oils over storage.

## Figures and Tables

**Figure 1 antioxidants-12-01231-f001:**
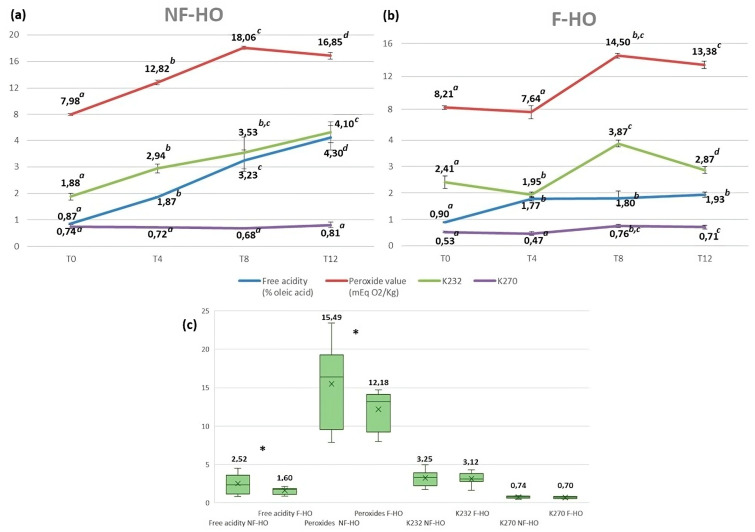
Evolution of free acidity, peroxide value (PV), K232, and K270 of non-filtered (NF-HO, (**a**)) and filtered (F-HO, (**b**)) cold-pressed hempseed oil during 12 weeks of storage in transparent glass bottles. Data are expressed as the mean ± sd of *n* = 3 oil bottles, each analyzed in triplicate. ^a–d^: different superscript letters in the same line indicate significantly different values for a given parameter (*p* < 0.05 by post hoc Tukey’s HSD test); the same superscript letters in the same line indicate not significantly different values (*p* > 0.05 by post hoc Tukey’s HSD test). Figure (**c**) illustrates the free acidity, PV, K232, and K270 of total NF-HO and F-HO samples. Data are expressed as the mean ± sd of *n* = 12 oil bottles, each analyzed in triplicate. In each box, “×” indicates the average value. In the comparison between NF-HO and F-HO samples, “*” indicates significantly different values (*p* < 0.05 by Student’s *t*-test) for a given parameter; conversely, the absence of “*” indicates non-significantly different values (*p* > 0.05 by Student’s *t*-test).

**Figure 2 antioxidants-12-01231-f002:**
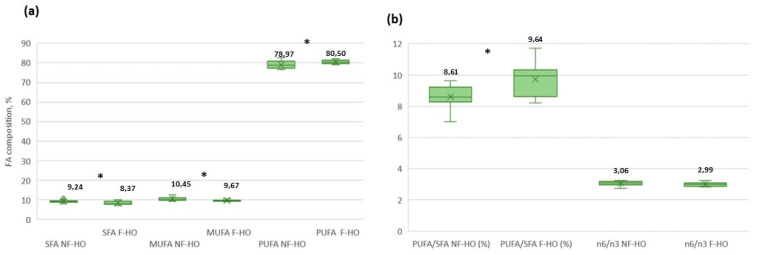
Main figures of the FA composition from total non-filtered (NF-HO) and filtered (F-HO) cold-pressed hempseed oil samples (**a**,**b**). Data are expressed as the mean ± sd of *n* = 12 oil bottles, each analyzed in triplicate. In every box, “×” indicates the average value, whereas the outlier points display the outlier data lying above the upper whisker. In the comparison between NF-HO and F-HO samples, “*” indicates significantly different values (*p* < 0.05 by Student’s *t*-test) for a given parameter; conversely, the absence of “*” indicates non-significantly different values (*p* > 0.05 by Student’s *t*-test).

**Figure 3 antioxidants-12-01231-f003:**
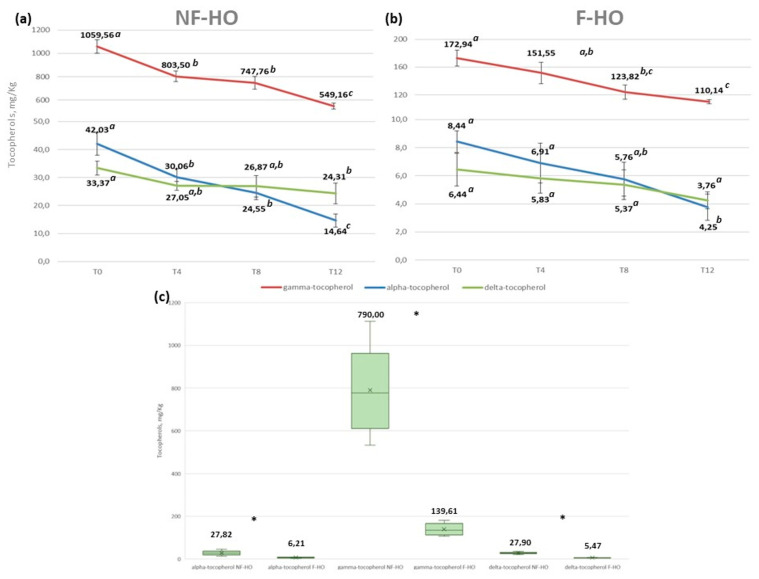
Evolution of the content of α- γ- and δ-tocopherol in non-filtered (NF–HO, (**a**)) and filtered (F–HO, (**b**)) cold-pressed hempseed oil during 12 weeks of storage in transparent glass bottles. Data are expressed as the mean ± sd of *n* = 3 oil bottles, each analyzed in triplicate. ^a–c^: different superscript letters in the same line indicate significantly different values for a given parameter (*p* < 0.05 by post hoc Tukey’s HSD test); the same superscript letters in the same line indicate not significantly different values (*p* > 0.05 by post hoc Tukey’s HSD test). Figure (**c**) illustrates the α- γ- and δ-tocopherol of total NF-HO and F-HO samples. Data are expressed as the mean ± sd of *n* = 12 oil bottles, each analyzed in triplicate. In each box, “×” indicates the average value. In the comparison of every analyte between NF-HO and F-HO samples, “*” indicates significantly different values (*p* < 0.05 by Student’s *t*-test).

**Figure 4 antioxidants-12-01231-f004:**
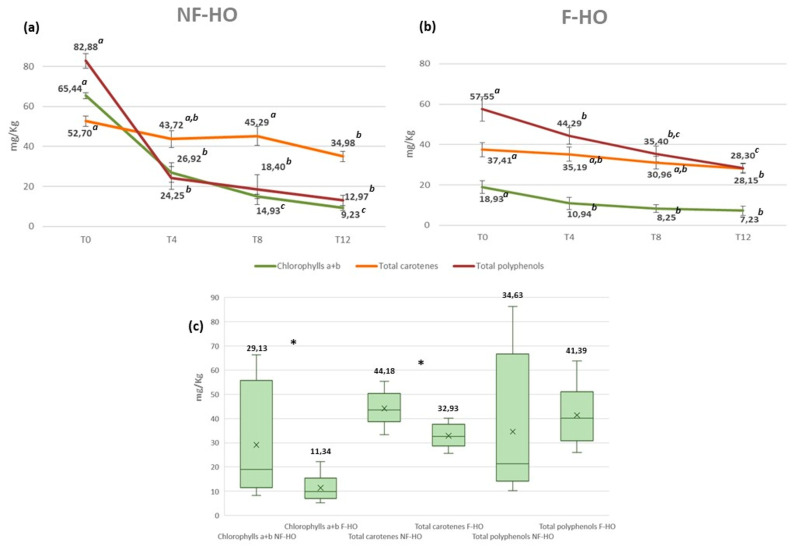
Evolution of the content of chlorophylls a + b, total carotenes, and total polyphenols in non-filtered (NF–HO, (**a**)) and filtered (F–HO, (**b**)) cold-pressed hempseed oil during 12 weeks of storage in transparent glass bottles. Data are expressed as the mean ± sd of *n* = 3 oil bottles, each analyzed in triplicate. ^a–c^: different superscript letters in the same line indicate significantly different values for a given parameter (*p* < 0.05 by post hoc Tukey’s HSD test); the same superscript letters in the same line indicate not significantly different values (*p* > 0.05 by post hoc Tukey’s HSD test). Figure (**c**) illustrates chlorophylls a + b, total carotenes, and total polyphenols of all NF-HO and F-HO samples. Data are expressed as mean ± sd of *n* = 12 oil bottles, each analyzed in triplicate. In each box, “×” indicates the average value. In the comparison of every analyte between NF-HO and F-HO samples, “*” indicates significantly different values (*p* < 0.05 by Student’s *t*-test); conversely, the absence of “*” indicates non-significantly different values (*p* > 0.05 by Student’s *t*-test).

**Figure 5 antioxidants-12-01231-f005:**
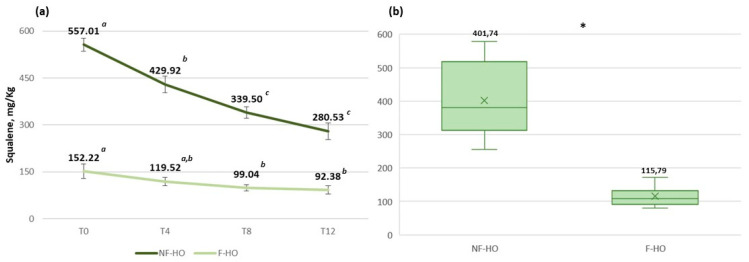
Evolution of the content of squalene in non-filtered (NF-HO) and filtered (F-HO) cold-pressed hempseed oil during 12 weeks of storage in transparent glass bottles (**a**). Data are expressed as the mean ± sd of *n* = 3 oil bottles, each analyzed in triplicate. ^a–c^: different superscript letters in the same line indicate significantly different values for a given parameter (*p* < 0.05 by post hoc Tukey’s HSD test); the same superscript letters in the same line indicate not significantly different values (*p* > 0.05 by post hoc Tukey’s HSD test). Figure (**b**) illustrates the mean content of squalene in all NF-HO and F-HO samples. Data are expressed as the mean ± sd of *n* = 12 oil bottles, each analyzed in triplicate. In each box, “×” indicates the average value. In the comparison of squalene between NF-HO and F-HO samples, “*” indicates significantly different values (*p* < 0.05 by Student’s *t*-test).

**Figure 6 antioxidants-12-01231-f006:**
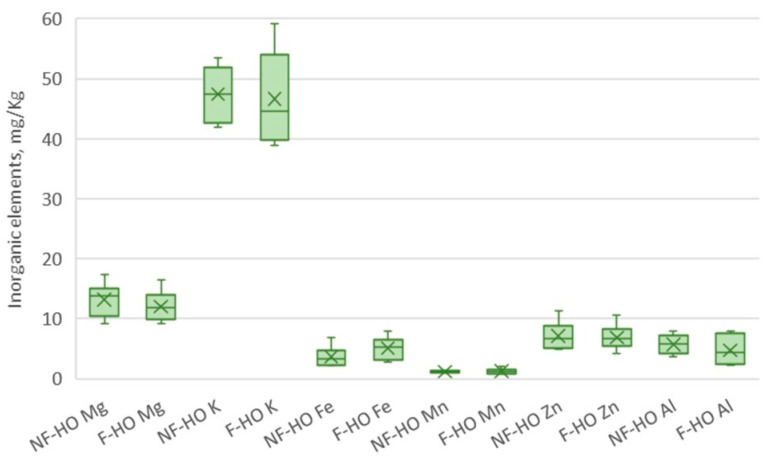
Mean content (mg/kg) of the most abundant elements present in total non-filtered (NF-HO) and filtered (F-HO) cold-pressed hempseed oil samples. Data are expressed as the mean ± sd of *n* = 12 oil bottles, each analyzed in triplicate. For each element, “×” indicates the average value. In the comparison of each element between NF-HO and F-HO samples, the absence of asterisk indicates non-significantly different values (*p* > 0.05, by Student’s *t*-test).

**Table 1 antioxidants-12-01231-t001:** Evolution of the FA composition (%) in non-filtered (NF-HO) and filtered (F-HO) cold-pressed hempseed oil during 12 weeks of storage in transparent glass bottles. Data are expressed as the mean ± sd of *n* = 3 oil bottles, each analyzed in triplicate.

	NF-HO				F-HO			
	T0	T4	T8	T12	T0	T4	T8	T12
**C16:0**	7.20 ± 0.45 ^a^	6.30 ± 0.44 ^a,b^	6.82 ± 0.33 ^a,b^	5.87 ± 0.31 ^b^	6.77 ± 0.23 ^a^	5.62 ± 0.53 ^b^	5.57 ± 0.32 ^b^	5.53 ± 0.19 ^b^
**C18:0**	3.24 ± 0.29 ^a^	2.94 ± 0.08 ^a^	2.18 ± 0.34 ^a^	2.40 ± 0.67 ^a^	2.95 ± 0.08 ^a^	2.58 ± 0.61 ^a^	2.18 ± 0.12 ^a^	2.27 ± 0.08 ^a^
** *SFA* **	*10.44 ± 0.74* ^a^	*9.24 ± 0.39* ^a,b^	*9.00 ± 0.03* ^b,c^	*8.27 ± 0.48* ^b,c^	*9.72* *±* *0.24* ^a^	*8.20* *±* *1.14* ^a,b^	*7.75* *±* *0.38* ^b^	*7.80* *±* *0.12* ^b^
**C18:1 n-9**	10.42 ± 1.08 ^a^	9.41 ± 1.03 ^a^	9.04 ± 0.23 ^a^	9.06 ± 0.67 ^a^	9.18 ± 0.33 ^a^	8.75 ± 0.21 ^a,b^	8.80 ± 0.33 ^a,b^	8.42 ± 0.19 ^b^
**C18:1 n-7**	0.93 ± 0.13 ^a^	1.13 ± 0.44 ^a^	0.96 ± 0.22 ^a^	0.87 ± 0.08 ^a^	1.00 ± 0.12 ^a^	0.88 ± 0.08 ^a^	0.81 ± 0.03 ^a^	0.87 ± 0.09 ^a^
** *MUFA* **	*11.35 ± 1.21* ^a^	*10.54 ± 1.46* ^a^	*10.01 ± 0.15* ^a^	*9.93 ± 0.74* ^a^	*10.18* *±* *0.42* ^a^	*9.63* *±* *0.13* ^a,b^	*9.61* *±* *0.35* ^a,b^	*9.29* *±* *0.21* ^b^
**C18:2 n-6**	55.56 ± 0.61 ^a^	55.18 ± 1.62 ^a,b^	52.74 ± 0.90 ^b,c^	52.36 ± 0.87 ^c^	56.27 ± 0.40 ^a^	56.13 ± 1.25 ^a^	54.61 ± 0.64 ^a^	54.29 ± 0.49 ^a^
**C18:3 n-6**	5.34 ± 0.95 ^a^	5.47 ± 0.46 ^a^	5.86 ± 0.37 ^a^	5.40 ± 0.83 ^a^	5.18 ± 0.36 ^a^	4.89 ± 0.14 ^a^	4.71 ± 0.17 ^a^	4.94 ± 0.06 ^a^
**C18:3 n-3**	18.85 ± 0.75 ^a^	17.92 ± 0.34 ^a^	18.37 ± 0.76 ^a^	18.70 ± 0.35 ^a^	19.18 ± 0.40 ^a^	18.63 ± 0.68 ^a^	18.80 ± 0.33 ^a^	19.07 ± 0.19 ^a^
**C18:4 n-3**	0.99 ± 0.12 ^a^	0.90 ± 0.21 ^a^	1.00 ± 0.23 ^a^	1.25 ± 0.38 ^a^	1.02 ± 0.13 ^a^	0.99 ± 0.24 ^a^	1.33 ± 0.34 ^a,b^	1.76 ± 0.11 ^b^
** *PUFA* **	*80.74 ± 2.20* ^a^	*79.47 ± 2.30* ^a^	*77.97 ± 1.38 * ^a^	*77.71 ± 1.44 * ^a^	*81.87* *±* *0.41* ^a^	*80.64* *±* *0.78* ^a,b^	*79.44* *±* *0.23* ^b^	*80.06* *±* *0.72* ^b^
** *PUFA/SFA* **	*7.77 ± 0.75 * ^a^	*8.62 ± 0.59 * ^a,b^	*8.67 ± 0.13 * ^a,b^	*9.41 ± 0.37 * ^b^	*8.43* *±* *0.18* ^a^	*9.98* *±* *1.54* ^a^	*10.27* *±* *0.55* ^a^	*10.27* *±* *0.09* ^a^
** *n-6/n-3* **	*3.07 ± 0.10 * ^a^	*3.22 ± 0.03 * ^a,b^	*3.03 ± 0.12 * ^a^	*2.90 ± 0.13 * ^a,c^	*3.05* *±* *0.05* ^a^	*3.11* *±* *0.13* ^a,b^	*2.95* *±* *0.13* ^a^	*2.84* *±* *0.02* ^a,c^

^a–c^: Different superscript letters in the same row indicate significantly different values for a given parameter (*p* < 0.05 by post hoc Tukey’s HSD test); the same superscript letters in the same line indicate not significantly different values (*p* > 0.05 by post hoc Tukey’s HSD test).

**Table 2 antioxidants-12-01231-t002:** Recent literature on the content (mg/kg) of α-, γ -, and δ-tocopherols revealed in non-treated and treated hempseed oils.

Sample	α-Tocopherol	γ-Tocopherol	δ-Tocopherol	Reference
**Commercial cold-pressed oils**	38.6–77.6	625.3–1013.2	14.0–35.1	[19]
**Commercial hot/cold-pressed oils**	20.8–74.8	99.9–576.7	16.6–49.1	[71]
**Array of filtered and non-filtered cold-pressed oils**	14.6–53.0	594–967	19.6–50.3	[42]
**Commercial cold-pressed oils**	21.02–65.92	376.28–906.63	-	[48]
**Commercial cold-pressed oils**	39.2–47.7	774.3–924.5	3.2–4.0	[72]
**Commercial cold-pressed oils subjected to cotton filtration and centrifugation in laboratory**	38.73	794.66	29.22	[29]

**Table 3 antioxidants-12-01231-t003:** Recent literature on the content (mg/kg) of chlorophylls a + b, total carotenes, and level (mg GAE/kg) of total polyphenols revealed in non-treated and treated hempseed oils.

Sample	Chlorophylls a + b	Total Carotenes	Total Polyphenols	Reference
**Commercial cold-pressed oils**	34.8–76.4	2.61–1.78	-	[19]
**Commercial refined oils**	0.41–2.64	0.29–1.73	22,100–160,800	[20]
**Commercial hot/cold-pressed oils**	-	2.37–52.15	8.32–200.42	[68]
**Commercial cold-pressed oils**	33.5–67.8	-	162.5	[26]
**Commercial cold-pressed oils subjected to ultrasound bleaching**	0.4–11.5	-	106.0–118.1	
**Commercial cold-pressed oils**	56.3	23.4	-	[23]
**Commercial cold-pressed oils** **subjected to ultrasound bleaching**	7.8–14.8	2.3–4.0	-	
**Array of filtered and non-filtered commercial cold-pressed oils**	0.78–75-7	2.53–3.93	-	[42]
**Commercial cold-pressed oils**	-	-	12.08–186.78	[29]
**Commercial cold-pressed oils**	-	-	290,320–384,520	[69]

**Table 4 antioxidants-12-01231-t004:** The recent literature on the content (mg/kg) of squalene revealed in hempseed oils.

Sample	Squalene	Reference
**Commercial oil**	ND	[93]
**Commercial hot/cold-pressed oils**	521.4–30,594.90	[68]
**Refined commercial oil**	80.52	[94]
**Commercial oil**	13.9	[95]

**Table 5 antioxidants-12-01231-t005:** Element profile (mg/kg) of non-filtered (NF-HO) and filtered (F-HO) cold-pressed hempseed oils before the experimental trial (T0) and after 12 weeks of storage in transparent glass bottles (T12). Data are expressed as the mean ± sd of *n* = 3 oil bottles, each analyzed in triplicate. Limit of Detection (LOD) of Se: 0.002 mg/kg.

	NF-HO		F-HO	
	T0	T12	T0	T12
**Na**	238.95 ± 17.54 ^a^	233.95 ± 21.91 ^a^	254.14 ± 6.71 ^a^	246.83 ± 13.30 ^a^
**Mg**	13.99 ± 3.22 ^a^	12.45 ± 2.76 ^a^	12.92 ± 3.71 ^a^	11.22 ± 1.11 ^a^
**K**	46.40 ± 4.67 ^a^	48.47 ± 5.25 ^a^	45.39 ± 6.31 ^a^	47.80 ± 10.28 ^a^
**Fe**	4.11 ± 2.36 ^a^	3.17 ± 0.89 ^a^	5.28 ± 2.54 ^a^	4.84 ± 1.42 ^a^
**Cu**	0.028 ± 0.026 ^a^	0.071 ± 0.035 ^a^	0.047 ± 0.025 ^a^	0.070 ± 0.019 ^a^
**Mn**	1.17 ± 0.20 ^a^	1.25 ± 0.19 ^a^	1.12 ± 0.23 ^a^	1.43 ± 0.58 ^a^
**Zn**	7.86 ± 3.28 ^a^	6.49 ± 1.46 ^a^	7.95 ± 2.42 ^a^	5.83 ± 1.50 ^a^
**Se**	<LOD	<LOD	<LOD	<LOD
**Ni**	0.023 ± 0.009 ^a^	0.014 ± 0.004 ^a^	0.016 ± 0.004 ^a^	0.020 ± 0.009 ^a^
**Cr**	0.008 ± 0.003 ^a^	0.006 ± 0.002 ^a^	0.006 ± 0.004 ^a^	0.007 ± 0.003 ^a^
**Al**	5.74 ± 2.15 ^a^	5.74 ± 1.28 ^a^	4.58 ± 2.60 ^a^	4.94 ± 2.88 ^a^
**Cd**	0.08 ± 0.004 ^a^	0.011 ± 0.005 ^a^	0.015 ± 0.004 ^a^	0.012 ± 0.006 ^a^
**Pb**	0.26 ± 0.08 ^a^	0.33 ± 0.07 ^a^	0.41 ± 0.24 ^a^	0.38 ± 0.14 ^a^
**As**	0.08 ± 0.004 ^a^	0.009 ± 0.001 ^a^	0.009 ± 0.003 ^a^	0.009 ± 0.007 ^a^

The same superscript letter “a” in the same line indicates not significantly different values (*p* > 0.05 by post hoc Tukey’s HSD test).

## Data Availability

Data are contained within the article.

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
