# Peer review of "Effect of Filtration Process on Oxidative Stability and Minor Compounds of the Cold-Pressed Hempseed Oil during Storage"

_antioxidants, 2023, doi:10.3390/antiox12061231_

Round 1
Reviewer 1 Report
In the article titled: "Effect of filtration process on oxidative stability and minor compounds of the cold-pressed hempseed oil during storage" there are too many flaws, so it cannot be published in its present form.
Below The Authors can find comments and recommendations of the Reviewer.
1. line 69 - "whose" is not correct here, sentence in lines 65-71 is too long and should be rewritten.
2. line 68 - it should be: "typical for" instead of: "typical of".
3. line 94 - reference is not complete and should be corrected.
4. line 94 - "to" is not needed here ("the improvement of oil processing").
5. 95-96 - references are not correctly presented.
6. line 121 - please provide more details about expeller press that was used.
Please explain, why such temperature of 50C was used for pressing the oil. Is it not too high?
7. line 153 - "reported in Cost and colleagues" is not correct, please rearrange the sentence.
8. line 157 - why acidity is expressed as % of oleic acid? Please explain.
9. line 168 - it should be PV, instead of peroxides
10. line 295 - "free fatty acids became higher" - the style is not correct, please rewrite the sentence.
11. line 310 - it should be: "in which" instead of: "where"
12. line 324 - "generally, the solubility of gas in oils decreases as temperature increases, exept for the case of oxygen" - please give comment about Your statement in comparison to cited here.
13. line 329 - "they grew" is not correct.
14. line 331 - "our" is not needed here.
15. line 340 - "varying" is gramatically not correct here, please rewrite the sentence.
16. line 343 - PV and kg should be used.
17. line 371 - it should be: "autooxidation".
18. line 380 - reference form is not correct.
19. line 444 - reference form is not correct; (Golimowski 2223?)
20. lines 456, 458, 466 - the year of publication is missing.
21. line 459 - the style of the sentence is incorrect, please rewrite it.
22. line 484 - it should be alpha-tocopherol.
23. line 486 - the sentence is incorrect.
34. line 505 - "had" is not a proper word here, please correct the sentence.
35. line 516, 520 - year is missing.
36. Tables 2, 3, 4 - "information" is not needed here.
37. Tables 2, 3, 4 - references are incorrect.
38. lines 534, 536, 552, 586, 600, 619, 648, 651, 657, 661, 666, 673, 675, 688, 690, 721, 728, 743, 766 - the references are not correct, please correct the references in the article according to "Instruction for Authors".
39. lines 572-573 - please explain, why such assessment could not be made.
They are essential for such study.
40. line 680 - information is missing.
41. line 793 - it should be: "content'
Moderate editing of English language is needed, according to comments presented in the review.
Author Response
RESPONSE TO REVIEWER 1
In the article titled: "Effect of filtration process on oxidative stability and minor compounds of the cold-pressed hempseed oil during storage" there are too many flaws, so it cannot be published in its present form.
Below The Authors can find comments and recommendations of the Reviewer.
- line 69 - "whose" is not correct here, sentence in lines 65-71 is too long and should be rewritten.
- line 68 - it should be: "typical for" instead of: "typical of".
R.:/ The period has been corrected and rearranged. Please see lines 61-68.
- line 94 - reference is not complete and should be corrected.
R.:/ References in the text and bibliography have been rearranged according to the formatting style of the journal.
- line 94 - "to" is not needed here ("the improvement of oil processing").
R.:/ It has been corrected
- 95-96 - references are not correctly presented.
R.:/ see the comment above
- line 121 - please provide more details about expeller press that was used.
Please explain, why such temperature of 50C was used for pressing the oil. Is it not too high?
R.:/ Please, see lines 112-121.
- line 153 - "reported in Cost and colleagues" is not correct, please rearrange the sentence.
R.: it is “reported in Costa and colleagues [33]”.
- line 157 - why acidity is expressed as % of oleic acid? Please explain.
R.:/ We have preferred to express the acidity in terms of % oleic acid to make easier the comparison of this cold-pressed oil with other very common edible oils, such as olive and seed oils. Additionally, literature on cold-pressed hempseed oil often reports acidity in terms of %oleic acid. Please see:
-Tura, M., Mandrioli, M., Valli, E., & Toschi, T. G. (2023). Quality indexes and composition of 13 commercial hemp seed oils. Journal of Food Composition and Analysis, 117, 105112.
-Rapa, M., Ciano, S., Rocchi, A., D’Ascenzo, F., Ruggieri, R., & Vinci, G. (2019). Hempseed oil quality parameters: Optimization of sustainable methods by miniaturization. Sustainability, 11(11), 3104.
-Izzo, L., Pacifico, S., Piccolella, S., Castaldo, L., Narváez, A., Grosso, M., & Ritieni, A. (2020). Chemical analysis of minor bioactive components and cannabidiolic acid in commercial hemp seed oil. Molecules, 25(16), 3710.
- line 168 - it should be PV, instead of peroxides
R.:/ It has been changed throughout the manuscript.
- line 295 - "free fatty acids became higher" - the style is not correct, please rewrite the sentence.
R.:/ It has been corrected.
- line 310 - it should be: "in which" instead of: "where"
R.:/ It has been corrected.
- line 324 - "generally, the solubility of gas in oils decreases as temperature increases, exept for the case of oxygen" - please give comment about Your statement in comparison to cited here.
R.:/ The solubility of gases in water generally decreases as temperature increases, following the Henry Law. This is not the case of gases in oil or lipid media, where the Henry's constant of water should not be used. Little and dated literature addressed oxygen solubility in oils and several limits on the design of the study or the choice of the measurement method have been pointed out. Recently, Cuvelier et al. (2017) have demonstrated that the solubility of oxygen in sunflower oil is 4/5 times higher than in water and increases with the temperature (in a range of 5-50°C). Nevertheless, the issue still needs to be taken “with a grain of salt”, because a variability related to the type of oil media (e.g., fruit or seed oils etc…), the choice of the temperature range, and the measurement method, is conceivable and requires further exploration. Recognizing the thoroughness with which the solubility of oxygen in oil in function of temperature has been approached, the period has been reformulated.
- line 329 - "they grew" is not correct.
R.:/ It has been corrected.
- line 331 - "our" is not needed here.
R.:/ It has been corrected.
- line 340 - "varying" is gramatically not correct here, please rewrite the sentence.
R.:/The period has been reformulated.
- line 343 - PV and kg should be used.
R.:/ They have been corrected throughout the manuscript.
- line 371 - it should be: "autooxidation".
R.:/ It has been corrected throughout the manuscript.
- line 380 - reference form is not correct.
- line 444 - reference form is not correct; (Golimowski 2223?)
- lines 456, 458, 466 - the year of publication is missing.
- line 459 - the style of the sentence is incorrect, please rewrite it.
R.:/ References in the text have been rearranged according to the formatting style of the journal.
- line 484 - it should be alpha-tocopherol.
R.:/ It has been corrected along with delta-tocopherol.
- line 486 - the sentence is incorrect.
R.:/ why? And precisely where?
- line 505 - "had" is not a proper word here, please correct the sentence.
R.:/ It has been corrected.
- line 516, 520 - year is missing.
R.:/ See comments 18-21
- Tables 2, 3, 4 - "information" is not needed here.
R.:/ “Information” has been removed from each table
- Tables 2, 3, 4 - references are incorrect.
R.:/ See comments 18-21
- lines 534, 536, 552, 586, 600, 619, 648, 651, 657, 661, 666, 673, 675, 688, 690, 721, 728, 743, 766 - the references are not correct, please correct the references in the article according to "Instruction for Authors".
R.:/ See comments 18-21
- lines 572-573 - please explain, why such assessment could not be made.
R.:/ a proper colour assessment could not be performed due to the lack of suitable instrumentation, such as a colorimeter capable of estimating colour in the CIELAB space.
They are essential for such study.
- line 680 - information is missing.
R.:/ It has been corrected.
- line 793 - it should be: "content'
R.:/ not clear what you are referring to.

Reviewer 2 Report
Manuscript antioxidants-2362085 presents an interesting and applicative study. Some comments:
- The quality of tables and figures must be thoroughly improved: improved images resolution; measurements units in all cases - tables and figures (not only in the caption); figures axes to be completely defined (parameters and measurement units).
- References are very difficult to follow, as different styles are used – text (name and year) and References section (numbered). To be corrected.
- The title refers to filtration. In the 2. Materials and Methods, a separate subsection should describe the filtration process – equipment, operation, parameters.
Author Response
RESPONSE TO REVIEWER 2
Manuscript antioxidants-2362085 presents an interesting and applicative study.
Some comments:
-The quality of tables and figures must be thoroughly improved: improved images resolution; measurements units in all cases - tables and figures (not only in the caption); figures axes to be completely defined (parameters and measurement units).
R.:/-For images, the original files were uploaded to the submission system at maximum resolution and they may not be displayed correctly in the manuscript with the current formatting.
-For the tables, they were created according to the predefined models provided by the manuscript template
-References are very difficult to follow, as different styles are used – text (name and year) and References section (numbered). To be corrected.
R.:/ References in the text and bibliography have been rearranged according to the Journal Style.
-The title refers to filtration. In the 2. Materials and Methods, a separate subsection should describe the filtration process – equipment, operation, parameters.
R.:/ A separate paragraph has been created to better explain oil production and filtration processes.

Reviewer 3 Report
I have now read the paper entitled Effect of Filtration Process on Oxidative Stability and Minor 2 Compounds of The Cold-Pressed Hempseed Oil During Stor- 3 age which is very actual by it subject. The hemp seed oil is a very high nutritional oil due in especially to it omega 3 content which is in optimum ratio with omega 6. Even if many oils has essential fatty acids not many of them are rich in omega 3 which seems to be a deficient in human consumption. On the market exist and other oils with high omega 3 content like flaxseed oil but their problem is from their sensorial characteristics. Hemp seed oil has a good taste and may be used and consumed very easily. The methodology used is very complex one and the conclusions are clear. The manuscript has some redactions problems (for example the bibliography which does not respect the MDPI standard journals) but this is not such a big problem (easily to be resolved). For me the box figures are not so easily to be fallowed but their change is optional. It would be interesting for consumers and industrial point of view the sensory characteristics of the oils filtered and non-filtered. Also their color changes through a colorimetric method.
Author Response
RESPONSE TO REVIEWER 3
I have now read the paper entitled Effect of Filtration Process on Oxidative Stability and Minor 2 Compounds of The Cold-Pressed Hempseed Oil During Storage which is very actual by it subject. The hemp seed oil is a very high nutritional oil due in especially to it omega 3 content which is in optimum ratio with omega 6. Even if many oils has essential fatty acids not many of them are rich in omega 3 which seems to be a deficient in human consumption. On the market exist and other oils with high omega 3 content like flaxseed oil but their problem is from their sensorial characteristics. Hemp seed oil has a good taste and may be used and consumed very easily. The methodology used is very complex one and the conclusions are clear. The manuscript has some redactions problems (for example the bibliography which does not respect the MDPI standard journals) but this is not such a big problem (easily to be resolved). For me the box figures are not so easily to be fallowed but their change is optional. It would be interesting for consumers and industrial point of view the sensory characteristics of the oils filtered and non-filtered. Also their color changes through a colorimetric method.
R.:/ thank you for the positive comments! We agree with the reviewer that the colorimetric and sensory analysis of these oils would have further enhanced the already robust study. However, oil samples from the study are no longer available and at the time of the study we lacked proper instrumentation as well as trained personnel to conduct this type of analyses.

Round 2
Reviewer 2 Report
The authors improved their work; however, I still recommend that figures axes to be completely defined (parameters and measurement units). For instance: in Figure 2, instead of “%”, it should be “oil composition, %”. Furthermore Fig. 2 should indicate a and b for the two graphs, with clear designation of a and b in the caption.
Author Response
The authors improved their work; however, I still recommend that figures axes to be completely defined (parameters and measurement units). For instance: in Figure 2, instead of “%”, it should be “oil composition, %”. Furthermore Fig. 2 should indicate a and b for the two graphs, with clear designation of a and b in the caption.
R:/ Dear Reviewer, with the exception of Figure 1 and 4 showing different types of analytes in the same graph, all the other figures have been changed in their axes, as you required.
